# Supporting Decentralised Energy Management through Smart Monitoring Systems in Public Authorities †

**Graeme Stuart \*** and **Leticia Ozawa-Meida**

Institute of Energy and Sustainable Development, De Montfort University, Leicester LE1 9BH, UK;
lozawa-meida@dmu.ac.uk

\* Correspondence: gstuart@dmu.ac.uk

† Interview and focus group data included in this paper were originally published in our paper presented at the 2019 ECEEE Summer Study: Bull R.; Dooley K.; Ozawa-Meida L.; Stuart G. In Sufficiently Engaged? How smart metering systems help local authorities become smart cities, ECEEE Summer Study, Belambra Presqu'île de Giens, France, 3–8 June 2019; European Council for an Energy-Efficient Economy.

**Abstract:** Energy infrastructure in large, multi-site organisations such as municipal authorities, is often heterogeneous in terms of factors such as age and complexity of the technology deployed. Responsibility for day-to-day operation and maintenance of this infrastructure is typically dispersed across large numbers of individuals and impacts on even larger numbers of building users. Yet, the diverse population of stakeholders with an interest in the operation and development of this dynamic infrastructure typically have little or no visibility of energy and water usage. This paper explores the integration of utility metering data into urban management processes via the deployment of an accessible "smart meter" monitoring system. The system is deployed in three public authorities and the impact of the system is investigated based on the triangulation of evidence from semi-structured interviews and case studies. The research is framed from three perspectives: the bottom-up micro-level (individual and local), the top-down macro-level (organisation-wide and strategic) and intermediate meso-level (community-focused and operation). Evidence shows that improved communication across these levels enables a decentralisation and joining-up of energy management. Evidence points to the importance of reducing the cognitive load associated with monitoring systems. Better access to information supports more local autonomy, easier communication and cooperation between stakeholders and fosters the conditions necessary for adaptive practices to emerge.

**Keywords:** smart monitoring; public buildings; communities of practice; decentralised energy management

## 1. Introduction

A large proportion of the world's population has moved to urban areas in the last 100 years and this trend continues. It is predicted that urban residents will reach around 6 billion by 2045, requiring basic services, such as energy, infrastructure, decent and affordable housing, etc. [1]. Despite cities only covering around 3% of the total world's area, they have a significant effect on climate change being responsible for two thirds of the global energy consumption and more than 70% of greenhouse gas (GHG) emissions [1]. The term 'smart city' emerged last decade as a utopian vision of urban integration and efficiency that can eventually result in GHG emissions reductions [2]. According to the Smarter 2030 Report 'ICT Solutions for 21st Century Challenges', the digital economy

could deliver approximately 12 GtCO$_2$e of emissions savings by 2030 as a result of smart logistics, grids and buildings [3].

This paper describes an application of the "internet of things" (IoT) in an "Urban analytics" [4] energy management monitoring system. The core concept of IoT is to automate monitoring without the need for human involvement and to gather data that allow our computers to know about our 'things' and greatly reduce waste, loss and cost [5]. In this case, the 'things' are utility meters and public buildings. Automated meter reading (AMR) technology allows utility meters (such as water, gas, heat or electricity meters) to generate data automatically and transfer them to a central database where they are associated with device identifiers and made available to monitoring systems. Energy usage is often considered invisible. Monitoring systems make energy and water usage visible to organisations and allow them to improve evidence-based decision-making. Hereafter, reference to energy management, services and infrastructure includes energy and/or water.

Public sector building stock is often heterogeneous in terms of factors such as age and complexity of the technology deployed and the innumerable things that can (and often do) go wrong in such complicated systems. Energy infrastructure includes components at various nested scales from the macro (e.g., buildings, district heating systems, electricity and gas networks) to the meso (e.g., windows, HVAC systems and building energy management systems) and the micro (e.g., light bulbs, thermostatic radiator valves and laptops). The infrastructure is dynamic, it grows as new buildings are constructed and when buildings are refurbished or extended. It is also constantly being maintained as equipment is reconfigured, repaired or upgraded somewhere in the organisation every day by a veritable army of maintainers (HVAC technicians, facility managers, installers, caretakers, etc.). Management of the infrastructure has a fundamental impact on energy and water usage and is therefore a core element of energy management activities.

Responsibility for day-to-day operation and maintenance of the building infrastructure is typically dispersed across large numbers of individuals and impacts on even larger numbers of users. Thus, the population of stakeholders with an interest in the operation and development of such a dynamic infrastructure is large and diverse. These stakeholder groups can be broadly sub-divided into (1) building users, (2) energy professionals—those responsible for maintaining and operating energy/water infrastructure and (3) strategic managers—those responsible for strategic management of infrastructure. Individuals can be members of more than one group.

Researchers have proposed an epidemiological approach to energy end-use demand studies [6], drawing a parallel with the complexity of biology. Energy management in public authorities can be viewed as a complex adaptive system using the framework developed by Axelrod and Cohen [7]. Stakeholders use a variety of behavioural and technical strategies in patterned interaction with each other and with infrastructure to achieve their goals. Different stakeholder groups have diverse strategic options and experience different pressures associated with acting to support sustainable energy policy at the individual and collective level. The implementation of energy monitoring systems in this context is a socio-technical activity which can manifest itself in complex ways depending on how it is implemented and how the generated data interact with governance schemes [8].

The role of public authority energy managers is complex, involving investigation, scenario building and development of interventions [9]. The primary opportunities for influencing the environmental and financial costs of delivering energy services are through technical, behavioural and systemic interventions. Technical interventions impact on the infrastructure itself or its configuration (e.g., replacement components or control settings). Behavioural interventions impact on stakeholders' behaviour to reduce waste (e.g., avoid lighting unoccupied rooms) and reorganise behaviour (e.g., holiday shutdown checklists). Systemic interventions may include both technical and behavioural change and are typically large-scale strategic transformations such as changing the way a department operates and/or moving into a new building.

The cost of energy efficiency interventions can range from major capital expenditure projects to low-cost or even no-cost measures such as adjusting existing control settings.

However, identifying opportunities to make investments in energy efficiency interventions requires an investment of time and effort which is classified as a transaction cost [10] of the intervention. To design an appropriate response and to coordinate any collective action necessary adds to these costs which are incurred before the costs of the intervention itself are factored in. For smaller interventions which may be very cheap to implement but costly to identify, the transaction costs can dominate the total cost. In these cases, the transaction costs are a major barrier to implementation. Monitoring systems reduce these transaction costs by automatically compiling salient information helping to identify and diagnose opportunities for intervention. Monitoring systems do not save energy or water directly, they improve the efficiency and effectiveness of energy management processes.

Energy saving behaviour is often as invisible as energy usage. If building users cannot see energy usage, then it is also not easy to see which behaviours and individuals are causing or mitigating wastage. High transaction costs provide a barrier to action at the individual level. The motivation for individuals to invest time and effort into developing energy or water saving behaviours is diminished further by the "free rider effect" [11] leading to a "tragedy of the commons" [12] situation where stakeholders allocate little to no attention to the issue. Monitoring systems are a kind of 'public good' in the context of the organisation, they introduce new information which has the potential to change the dynamics of the system dramatically if the information is made available in a way that can be readily used. Transparency of performance measures such as energy consumption and success criteria such as reduced consumption can drive the adaptive selection of strategies of stakeholders within and between buildings.

The aim of this paper is to explore two research questions: (1) how can utility data be integrated into urban management processes through the deployment of smart meter monitoring systems? and (2) what are the impacts of implementing a smart, urban analytics system on the dynamics of energy and water management in public authorities? The research is framed from three perspectives within large, multi-site, public sector organisations: the bottom-up micro-level (individual and local), the top-down macro-level (organisation-wide and strategic) and the intermediate meso-level (community-focused and operational), similar to the approaches to low carbon transitions in energy demand suggested by Janda and Parag [13].

Two relevant components of smart monitoring systems relate to the capability of using metered data in energy management and how these data can support management and engagement with a wide range of stakeholders through energy feedback (described in Sections 1.1 and 1.2). Section 2 provides an overview of the urban analytics platform developed in the EU-funded Energy Data Innovation Network (EDI-Net) project. Evidence was elicited through interviews, focus groups and case studies following the methodology described in Section 3. Section 4 explains the institutional and social effects resulting from the use of the EDI-Net system in three European public authorities, followed by a discussion of the implications in energy management and stakeholder engagement in Section 5. Conclusions are presented in Section 6.

*1.1. Metering Data in Energy Management*

Monitoring systems traditionally provide technical information to energy professionals. Since they may be responsible for hundreds of buildings; energy management teams cannot easily conduct regular visits for energy surveys and audits. In many cases when a problem occurs (e.g., a control system failure or a water leak) it is not visible to building users. The longer such problems continue without being addressed, the greater the economic and environmental costs. There is potential for such problems to go unaddressed for months without a monitoring system in place.

Metered energy and water data have traditionally been a source of information that allows energy management teams to monitor usage remotely. Guidance on data analysis approaches for energy management has been published in the UK under The Energy Efficiency Best Practice Programme (EEBPP) since the 1990s [14]. These guides formalise and standardise techniques such as simple regression analysis with degree days or CUSUM (cumulative sum) analysis used by energy

professionals working with low-resolution, monthly data to establish 'normal' patterns of usage and identify divergence from these patterns. The point of this data analysis is to trigger investigation. The energy management function in an organisation can allocate resources to an exploratory mode of problem verification and diagnosis which ultimately leads to interventions being designed and implemented. Data can be used to identify the nature of a problem, to pinpoint when it began and to verify the effectiveness of any attempts to mitigate the problem.

Modern smart meter technology can monitor thousands of buildings at high resolution. A standard utility meter in the UK generates a reading every thirty minutes, producing 17,520 data points in a normal year (compared to 12 data points per year in a monthly monitoring system). With smart meters being rolled-out across the EU [15,16], it is not unusual for these data to be available to the central energy management function of large public authorities. When many complex buildings are being monitored in this way, the value of a data-driven approach is clear.

Data analysis approaches using monthly data have been adapted for use with higher resolution data from smart meters [17] and are still commonly used in industry [18]. Inverse modelling refers to a statistical model which can be 'fitted' using a method such as ordinary least squares (OLS) to measured data. The model fitting process estimates model parameters that can then be used to predict dependent variables based on the values of independent variables (e.g., to predict gas usage for a given building based on outside air temperature). Inverse modelling approaches such as PRISM [19] and the Inverse Modelling Toolkit [20] allow for a highly structured and reproducible method for generating predictive energy usage models based on historical usage and weather data.

These models allow for a hypothesis testing approach where the null hypothesis is that a building is continuing to operate as 'normal' and that no special attention is required. By observing the pattern of consumption relative to the forecast produced by an inverse model it is possible to reject the null hypothesis if consumption patterns diverge in a statistically significant way. Such tools convert data into usable information.

## 1.2. Energy Feedback and Visualisation

Previous research has identified the effectiveness of energy feedback in changing behaviour by 'making energy visible' [21–23]. In non-domestic buildings, energy consumption and its effects are largely invisible to building users if the space is comfortable, and equipment is working [24]. In organisational settings, building users do not have direct financial incentives to reduce energy use as they do at home because they do not have to pay energy bills and they rarely have access to information regarding their consumption levels and patterns (relative to previous periods) [25].

In operation, monitoring systems can provide non-technical users with on-demand access to up-to-date and reliable information relating to the performance of energy and water infrastructure in a form that is easy to absorb and is directly tied to objective measured data. This leads to more transparency and more opportunities for collaboration on technical and behavioural interventions for energy and water efficiency.

A closed feedback loop is achieved when information from a monitoring system is made available to stakeholders in a way that can influence their actions. When these actions impact on building performance and are fed back to the user via the monitoring system, the information is transformed into embodied knowledge. This kind of feedback loop can be established for building users only if the monitoring system is sensitive enough to detect the impact of the actions taken. Two critical aspects of a robust feedback loop are the effectiveness of the information uptake by users and its conversion into action. Energy data are transformed into information and presented in a form that is easy to absorb by the target audience. The selection and visualisation [26,27] of information will determine how accessible the system is and how persuasively it communicates to the user. Based on the clarity of the information provided (e.g., historical versus normative comparisons) [28–30] and the ability of the user to process this information, the communication can prompt immediate thoughtful behavioural choices or long-term cognitive engagement and enduring behaviour change [31–33].

Participating in a robust closed loop can enable building users and energy professionals to gain knowledge about how changes in their behaviour influence the building, to verify the effectiveness of their intentional actions and to observe otherwise invisible changes to the building performance. By using monitoring and feedback systems, highly engaged users can act as peer educators disseminating their knowledge on energy saving opportunities to colleagues [25]. Formal and informal networks (communities of practice) can emerge introducing and diffusing new models, concepts and practices, so these can become part of the organisation's culture [34]. Social norms can be enhanced when users perceive competition of energy performance improvement between buildings [35]. Competitive approaches (energy reduction competitions) and incentivisation models (gamification) can help to intensify knowledge exchange and participation among building users as well as promoting cooperative behaviour [36]. In this way, embodied knowledge can be upgraded to a kind of collective wisdom that can anticipate issues and choose the correct actions instinctively.

## 2. The Energy Data Innovation Network

The three-year (2016–2019) EU-funded (Horizon 2020) EDI-Net project aimed to increase the capacity of European public authorities to use sub-hourly smart energy and water meter data to accelerate the implementation of sustainable energy policy, by introducing a step change in the quality and availability of building energy performance information. The EDI-Net system, developed in the project, was deployed in over 1300 buildings across three public authorities in Europe: Leicester (UK), Nuremberg (DE) and Generalitat of Catalonia (ES). The EDI-Net system was deployed in 1,056 buildings in Generalitat of Catalonia, 236 buildings in Leicester and 37 buildings in the city of Nuremberg during the project lifetime.

The EDI-Net system comprises three main tools: the dashboard, the benchmarking tool and the online forum. These systems were designed to meet three specific user requirements defined by the project: (1) to allow stakeholders to track energy performance and communicate this performance in a user-friendly way (dashboard), (2) to facilitate communication between stakeholders (online forum) and (3) to manage intervention plans for energy efficiency (benchmarking tool). In Leicester and Nuremberg, the dashboard was used for energy monitoring and for communication between stakeholders in mobilisation campaigns. In Catalonia and Leicester, the benchmarking tool was used was used for planning energy-related investments and managing intervention plans for energy efficiency.

The dashboard and the benchmarking tool are based on calculations conducted by a common data analytics engine. The modelling approach is described in detail in Stuart and Fleming [37] The core of the calculation is an inverse model and a unitless performance indicator. The algorithm [37] was originally developed from a prototype designed in the SmartSpaces EU project [35]. Data are continually imported into the system, cleaned and mapped against local weather data drawn from open sources [38]. For each building, the system estimates what is 'normal' usage (gas, water, electricity, heat) from historic data. Each week, a new baseline model is fitted to the latest 12-months of data and used to generate a prediction of what usage would be expected if the building was behaving as it was during the baseline period. In order to reflect the uncertainty of the baseline model, the model residuals (errors) are analysed to give percentile values (10th, 25th, 75th and 90th) that, when added to the model prediction represent the 'normal' expected range.

### 2.1. EDI-Net Dashboard

Figure 1 shows the diagnostic report provided by the EDI-Net dashboard as a tool for expert users (though it is available for all users). Actual measured usage is shown as a black line and the baseline model predictions are represented as coloured zones. When interpreting these reports, the key thing to note is where the black line falls relative to the coloured zones. If the black line is within the yellow 'normal' zone, then the building is operating as expected. If the black line persistently falls within or goes beyond the red or green zones, then there may be a problem worth investigating. With practice, this can be interpreted intuitively.

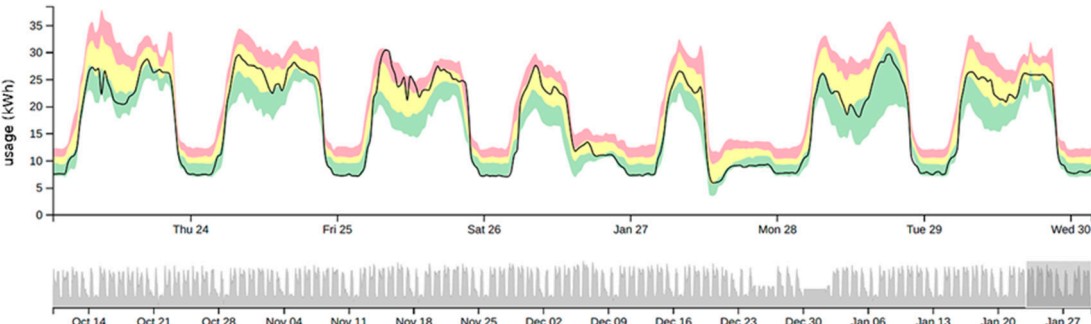

**Figure 1.** One week of usage (black line) shown with coloured 'zones'.

To present the data in an even more user-friendly way, the data are reduced to a unitless performance indicator. Each half-hourly value is compared to the model prediction and the percentile score of the difference is calculated with respect to the baseline model residuals. Details of the calculations can be found in Stuart and Fleming [37].

The resulting score is unitless and bounded so is easily converted into a simple visualisation of user-friendly smiley faces. A score of zero represents usage below any experienced during the baseline period and is visualised as a green, happy face. A score of one represents usage above any experienced during the baseline period and is visualised as a red, sad face. A value of 0.5 represents the median expected usage and is visualised as a yellow, neutral face. In between, the face colour and features vary gradually so any value can be expressed. In addition, the indicator can be aggregated by averaging over time, so it is useful for generating daily and weekly summaries.

Figure 2 shows the top-level league table interface. League tables show a list of similar buildings (e.g., all primary schools) sorted according to the average value of the performance indicator over the latest 7 days. Each item shows the building name against smiley faces for each utility plus a larger face calculated by averaging the utilities (i.e., gas, water and electricity). This simple format provides users with an overview of current performance across their entire building portfolio and highlights buildings where consumption is higher or lower than past performance suggests is normal. Therefore, any building can move from the bottom to the top of the list by making realistic changes. If buildings are operating normally, all will show yellow, neutral faces.

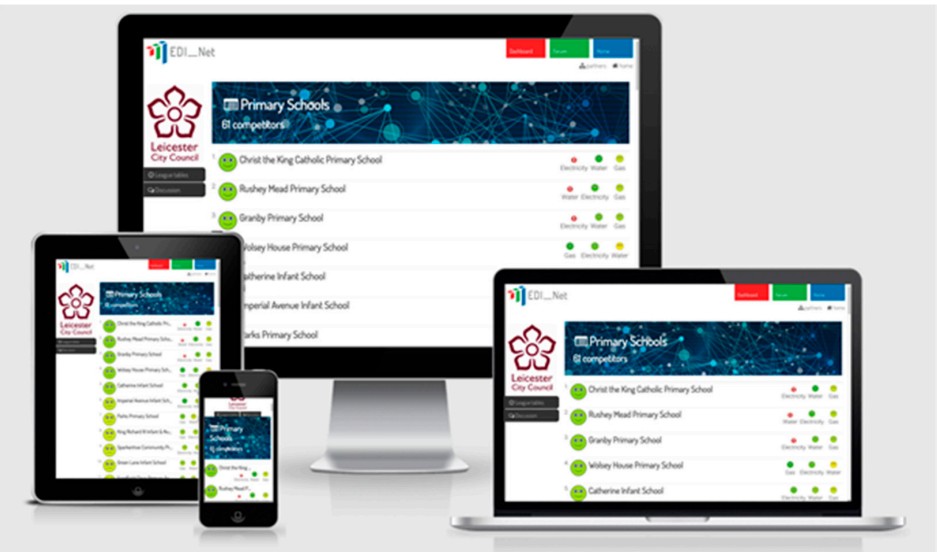

**Figure 2.** Screenshots of the EU-funded Energy Data Innovation Network (EDI-Net) dashboard league tables.

In addition to the dashboard, the EDI-Net online forum allows participants to share their experiences, promote their successes and discuss their challenges.

## 2.2. EDI-Net Benchmarking Tool

The EDI-Net benchmarking tool provides a more strategic view of the energy consumed over time in a building portfolio. The tool records detailed building data as well as technical and systemic intervention data. The user interface provides a set of features for management and intervention planning, including key contextual factors (building typology, use typology, building location, inside or cross-organisation), baseline analysis, a list of existing interventions and associated investment data and future recommended measures based on data analytics. A detailed explanation of the data-driven approach and algorithms used in the benchmarking tool can be found in Grillone et al. [39].

## 3. Materials and Methods

Quantitative and qualitative evidence was gathered throughout the project to explore how the EDI-Net system was integrated to the operational processes of the public authorities and its impact on their organisations. The overall approach is of methodological and data triangulation, including evidence from interviews and focus groups with different users (energy managers, decision makers and building users) as well as case studies drawn from across the three public authorities [40]. This approach was designed to evaluate the effectiveness of the EDI-Net system to support energy management and user engagement.

## 3.1. Interviews and Focus Groups

Interviews and focus groups (in semi-structured format) were conducted to obtain in-depth insights into the participants' thoughts, viewpoints, attitudes and actions [2]. These explored how the participants used the EDI-Net tools for energy management and for communicating with building users in their organisations to achieve a better understanding on how utility data can be integrated into urban management processes. Questions also sought to understand whether participants perceive that levels of awareness and knowledge of energy use were influenced at the individual or collective level and the ability of the system to create meso-level communities of practice as a result of viewing or interacting with the tools. Ethical approval was obtained via De Montfort University's review system in August 2016, with protocols observed to ensure participation was voluntary and participants were assured of anonymity.

A total of 28 participants were interviewed face-to-face, in focus groups or virtually via video conference between February and July 2018. Four interviews took place in Leicester in June–July 2018. Two virtual interviews were held in Nuremberg in July 2018. Two focus groups were held in Catalonia, an initial one in February 2018 with 12 energy/sustainability professionals (central managers) responsible for various buildings in their departments across the region, and a second meeting conducted in May 2018 for training purposes and to follow up issues that emerged in the first focus group. Twenty-two participants attended this second meeting (including the original 12) from the same departments. After introductions, interviews and focus groups lasted an average of 60 min and were recorded with the participants' authorisation.

Participants in interviews and focus groups were all directly involved in the implementation and communication of the EDI-Net services in their respective public authorities. Participants included energy coaches and energy champions, central or local energy managers and managers with decision-making roles related to energy efficiency investments. Table 1 summarises the participants roles and responsibilities.

**Table 1.** Summary of participants in the partner public authorities [2].

| Interviewees | Roles and Responsibilities |
|---|---|
| CAT-FG1 12 participants | • Central energy/sustainability managers responsible for various buildings in their departments across the Catalan region (mainly office buildings, but can also include museums, fire/police stations). <br>• Representatives of the following departments attended this focus group: Culture, Justice, Governance (3), Agriculture (2), Interior, Work, Presidency and the EDI-Net delivery manager. <br>• Central energy managers may also have energy-related finance responsibilities. |
| CAT-FG2 22 participants | • Including the 12 participants of the first focus group. <br>• Selected local energy managers from departments across the region responsible of various buildings. |
| LE1 | • Local site manager of a primary school in Leicester <br>• Energy champion, responsible to coordinate pupils in the Eco-Schools initiative |
| LE2 | • Local business manager of a secondary school in Leicester <br>• Energy champion, responsible to coordinate pupils in the Eco-Schools initiative <br>• Financial responsibility on energy efficiency investments |
| LE3 | • Energy and sustainability officer of a University in Leicester <br>• Energy champion, responsible of staff and student engagement |
| LE4 | • Energy manager of a University in Leicester <br>• Financial responsibility on energy efficiency investments |
| NUR1 | • Head of Municipal Energy Management in the city of Nuremberg <br>• Responsible of a portfolio of around 800 buildings <br>• Financial responsibility on energy efficiency investments |
| NUR2 | • Eco-teacher in a primary school in Nuremberg, responsible to teach energy in the curriculum (various ages) <br>• Energy champion, engaging staff and students |

The focus group sessions and all interviews were transcribed verbatim. Transcripts were initially coded using qualitative analysis software (NVivo version 11) according to the extent to which they aligned with the pre-identified themes of the EDI-Net evaluation framework. Interviews' content was broken down into tentative labels using open coding. Subsequently, axial coding was used to seek commonalities among the coded data and connections between emerged categories. Preliminary findings were presented to representatives of the public authorities to validate results or add further information.

*3.2. Case Studies*

The purpose of the case studies was to identify specific instances where reduced wastage, improved awareness or investment were achieved and attributable to the EDI-Net services from the perspective of the representatives in each public authority. Delivery managers were asked to select and describe best practice examples in their organisations that explain the benefits of the tools for solving energy management problems, engaging with different stakeholders and making energy use more visible. The case studies provided by the delivery managers described the background of the problem (evidence from the EDI-Net tools), actions taken, corresponding results of actions taken, financial information about the energy savings or investments and the potential for replication. In each case, there is evidence that the EDI-Net system supported decision-making and helped to optimise the overall energy management function. A full description of the case studies can be found in the EDI-Net project Final Evaluation Report [41].

**4. Results**

The gathered evidence demonstrated clearly how users interacted with the EDI-Net system and provided a qualitative and quantitative picture of the successful pathways to concrete organisational changes. Evidence elicited from interviews and focus groups identified institutional and social effects resulting from the use of its tools. The case studies indicated that avoided wastages or identified energy

efficiency investments can be directly attributed to the use of the EDI-Net tools. Sometimes energy/water wastage has a remarkably high cost and so the speed of response has a significant impact on the overall cost of the fault. The effect of the EDI-Net system in these cases is to highlight faults as soon as they occur.

*4.1. Institutional Effects*

The deployment of the EDI-Net system in the three public authorities can be regarded as communication-based systemic interventions. Institutions may respond to these interventions at the structural or policy level or by reducing barriers to change. Organisational impacts were observed in interviews and case studies, including the gradual decentralisation of energy management, the empowerment of local building users to monitor their consumption and decide actions based upon the observed consumption patterns, the ability to plan potential technical interventions and to support the implementation of sustainable energy policy.

4.1.1. Decentralisation of Energy Management

Energy and sustainability managers find the dashboard very useful to have a "quick overview of the energy consumption in each building" (NUR1) and an understanding of "how the buildings are performing" (LE3). One of the main features of the dashboard and benchmarking tool mentioned by interviewees was the ability to detect abnormal energy or water usage in a timely manner LE2, NUR1). As soon as the red faces were discovered by central or local energy managers in the municipalities, they communicated with the staff in the buildings to initiate countermeasures.

Case studies in Nuremberg documented that wastages of 12,800 $m^3$ of water (accounting for €50,000 of water costs) were avoided in cultural and school buildings through the identification of water leakages from the visualisations in the EDI-Net dashboard (Table 2). Central energy managers quickly detected malfunctions when the smiley face for the water consumption in these buildings turned deeply red. Notifications from central energy managers to local managers prompted on-site investigations. In the cultural building (CS1) (Table 2), the source of the problem was a recently maintained sprinkler system. After the notification, the building manager was able to partially switch off the sprinkler system (without shutting it completely), stopping it from wasting more water. In one of the schools (CS3) (Table 2), the water wastage was due to a defective valve of an automatic toilet flush which caused a continuous water flow in the toilets of the whole building. The valve was closed manually and later replaced. The wastage, which typically may not have been detected for weeks, was detected immediately and the problem was isolated and resolved in only 2 days. In another school (CS2) (Table 2), the excess water consumption was caused by a defective water system outside the facilities showing a constant high level in the detail graphs of the dashboard. This type of malfunction is not easily detectable for building operators on site. Only one day after the water wastage was detected and the janitor of the school identified the problem, a specialised plumber company repaired the malfunction. Timely notifications and quick actions to solve the malfunctions promptly avoiding water wastage that otherwise would have remained undetected during summer holidays or until the next annual maintenance of the equipment.

Similar cases were also reported in the other public authorities. In Catalonia, the timely detection of a malfunctioning pressure regulator was detected reducing the efficiency of the air conditioning system and overloading the compressor in a building of the Health Department indicating an overconsumption of 168 kWh per day (CS4) (Table 2). The repair avoided serious damage to the compressor and electricity wastage. In Leicester, excessive electricity and water usage were also quickly identified and corrected in a Transport Depot (CS5) (Table 2). Following corrective actions, the estimated electricity savings were around 35,916 kWh per year and water savings of around 263 $m^3$ per year.

**Table 2.** Case studies related to wastage detection.

| Case Study | Public Authority | Building Type | Commodity | Savings | Monetary Savings |
|---|---|---|---|---|---|
| CS1: Supporting on site managers | Nuremberg | Cultural | Water | 7600 m³ | €30,000 |
| CS2: Finding malfunctions that are not obvious | Nuremberg | School | Water | 2200 m³ | €8000 |
| CS3: Controlling a building when nobody is there to look at it | Nuremberg | School | Water | 3000 m³ | €12,000 |
| CS4: Detecting malfunctions with EDI-Net | Catalonia | Health | Electricity | 61,320 kWh | €1300 |
| CS5: Dashboard Exception Reporting | Leicester | Transport depot | Electricity Water | 35,916 kWh 263 m³ | €4937 [1] €750 |

[1] Exchange rate (January 2019): 1GBP = €1.15.

In Nuremberg, central energy managers typically conduct the energy monitoring, and contact local building managers when actions are required. Through the dashboard, energy management is gradually being decentralised. In schools, technical staff were less engaged than non-technical users as they considered energy monitoring to be the responsibility of the central energy management teams rather than their own jobs. However, this attitude gradually changed as the building managers responsible for schools committed to monitor the dashboard and give more responsibility to the local technical staff. Decentralisation of energy management empowers local building users to monitor their consumption and decide actions based on the consumption patterns. In turn, this frees up additional time for the central energy managers to carry out more complex efficiency initiatives and projects.

Similarly, schools in Leicester have also benefited local building users to manage their energy consumption, for example, to identify energy and water usage during school holidays and the ability to plan heating shut-downs or switch off campaigns (LE1, LE2). With current constrained budgets in the UK, schools are always interested in energy savings, but complex monitoring tools are difficult to manage for staff with busy day-jobs. The dashboard has empowered users to understand their consumption and react accordingly. Over 40 schools are using the dashboard on a regular basis. As a result, schools are now looking at ways to improve their performance on a daily and weekly basis.

#### 4.1.2. Managing Intervention Plans for Energy Efficiency Investments

The main features of the benchmarking tool that several focus group participants in Catalonia and one interviewee in Leicester found useful were the ability to document and monitor technical interventions, understand their economic and energy impacts and plan the potential implementation of further interventions (CAT-FG1, LE4). Other features that participants of the focus groups found useful were the provision of estimated annual savings, comparison of buildings to conduct a general analysis at the departmental or public authority level, documentation of energy efficiency measures, ability to learn from interventions conducted in other buildings and the capacity to follow up implemented measures as well as their costs.

The Generalitat of Catalonia established a specific organisational structure to manage energy efficiency interventions across different departments, using the benchmarking tool as their main means for comparison and analysis. One of the main problems was the difficulty to process and collect data of their large building portfolio, as the energy data were dispersed across different departments. The delivery manager in Catalonia explained that the tool had been deployed mainly among departmental energy managers responsible for energy and sustainability aspects in their departments and for their associated buildings (CAT-FG1). These managers input details of technical interventions into the tool.

At the public authority level, the delivery manager in Catalonia can visualise all 1162 buildings in the benchmarking tool. She explained that the tool has been used to conduct a general analysis at the public authority level to know the energy consumption by different building types and also by departments, and to follow up the interventions that have been implemented (CAT-FG1). Around 1911

interventions were implemented with an investment of €17,107,173 as part of Catalonia's regional strategy for energy renovation in buildings resulting in around 1.6% electricity savings (CS6) (Table 3).

**Table 3.** Case studies related to energy efficiency investments.

| Case Study | Public Authority | Buildings Involved | Technical Interventions | Investment | Electricity Savings |
|---|---|---|---|---|---|
| CS6: Relevant tool for regional policy | Catalonia | 1162 | 1911 | €17,107,173 | 4,876,356 kWh |
| CS7: Prioritising energy efficiency actions | Catalonia | 183 | 13 | €380,000 | N.A.[1] |
| CS8: Applying energy efficiency measures | Catalonia | 1 | 1 | €27,119 | 12,587 kWh |
| CS9: Lighting upgrade | Leicester | 22 | 22 | N.A.[1] | 1,595,905 kWh |

[1] N.A. Not available.

At the departmental level, the benchmarking tool was used to evaluate the performance and prioritise technical interventions in 183 buildings of the same typology (police headquarters) in the Department of Internal Affairs in Catalonia (CS7) (Table 3). With an annual available budget for energy renovations of €380,000 in 2018, the central energy manager had to allocate the annual budget for energy efficiency improvement of the building stock in the most cost-efficient manner to achieve the highest savings. Based on the benchmarking analysis of 18 buildings with the lowest performance, the budget was allocated to 13 buildings that showed an estimated improvement of about 25% of the cost-benefit ratio with respect to the investment allocation previous to EDI-Net.

In a similar manner, the benchmarking tool identified an inefficient building (Fire Brigade Office) with high electricity consumption during the heating and cooling seasons (CS8) (Table 2). Following an analysis of the energy performance of the building and recommended energy efficiency measures, an old inefficient air conditioning system was replaced by a geothermal system. This replacement is predicted to reduce energy and costs by around 33%.

In Leicester, the benchmarking tool has helped to accelerate the implementation of systemic interventions by allowing decision makers to easily quantify the impact of investment (CS7) (Table 3). Traditionally, Leicester City Council has used LED on a small scale. The benchmarking tool was used to document LED lighting upgrades in 5 council buildings including investment data as well as the energy, financial and carbon savings. The visual information of quantified savings in the benchmarking tool clearly showed the benefits of investing to decision-makers and secured the necessary funding to upgrade the lighting to LEDs in 22 operational buildings across the portfolio.

### 4.1.3. Support for Developing Sustainable Energy Policy

In Catalonia, the benchmarking tool has been used to support regional policy through understanding the impact of implemented technical interventions, allowing the public authority to generate useful knowledge for investment decisions in its large building portfolio and to improve investment decision-making for new energy renovation projects (CS6) (Table 3).

In contrast, the Head of Municipal Energy in the city of Nuremberg explained that because several technical interventions have already been implemented, behavioural interventions using the dashboard are becoming more relevant. The interviewee highlighted the importance of focusing on engagement in the future, particularly to reach the less aware and engaged: "the smiley faces are useful to inform and engage with building users, particularly to make energy visible . . . nobody has really been confronted with any energy at all, nobody thinks about energy at all . . . the smiley faces help people just to visualise that there is energy, and there is energy consumption" (NUR1). Based on the experience gained in the EDI-Net project, the city of Nuremberg would like to continue using the dashboard to further develop their public engagement strategies.

At the organisational level, one interviewee considered that raising awareness is also a relevant part of the internal energy policy, but the energy visualisation in screens or PCs needs to be complemented with the environmental champions' scheme and engagement campaigns (LE4). The interviewee

believes that the commitment of champions is necessary to support local actions in the buildings, while the campaigns help users to understand how their behaviour can affect the smiley faces.

*4.2. Social Effects*

Among the most important aspects of engaging people in collective action are the formal and informal networks that introduce and diffuse new practices into the organisation's culture. Evidence from interviews and case studies indicated the strengthening of internal and external networks through an enhanced communication of the energy/water performance of the buildings with energy management teams and users. Direct communication and discussion of energy performance of the buildings can result in increased cooperation and engagement among users to decrease excessive energy consumption through the creation of communities of practice or through competitive approaches.

Activities in schools are relevant for social effects to increase awareness and knowledge of energy efficiency across different communities. To support engagement with building users, EDI-Net was introduced in the Eco-Schools initiative in Leicester City Council and the "Keep Energy in Mind" (KEiM) programme in the city of Nuremberg. Teachers welcome tools that can be used in the classrooms to increase environmental awareness with their students. The freely available web-based dashboard and league tables are easily accessed by teachers and pupils keeping energy savings at the forefront of everybody's minds and facilitating discussion about the performance of their buildings.

4.2.1. Communicating with Stakeholders

Interviewees expressed that the communication of the dashboard and benchmarking tool to technical and non-technical users is very powerful. In the dashboard people can "instinctively" interpret the faces and get a sense of how urgent actions are needed: "if it is bright red, I know it is quite seriously unhappy ... and I need to find out what is going on" (LE4). In schools, the dashboard's simplicity (both visually and process-wise) was seen by staff and pupils as a great advantage. The tool was found to be easy to use and to understand even for the very-young students in infant schools (ages 4+). In one primary school in Leicester, the interviewee considered that the "smiley and sad faces communicate better than words" (LE1). The interviewee explained that when he asks pupils about the faces, the children can easily interpret that a sad face means that the school is not performing well. One teacher in Nuremberg mentioned that she uses the smiley faces with young children to highlight how the school uses heat and electricity, while she asks older pupils to evaluate the energy performance of the building through the detailed graphs (NUR2). This teacher also commented that she uses the tool to engage with colleagues and pupils and tell them to "switch off lights" or "do not use the heater if you do not need it" (NUR2).

At the organisational level, focus group participants in Catalonia agreed that the benchmarking tool has raised awareness in their periodic training activities. The visibility of the energy feedback has prompted local energy teams to become more proactive in various departments, and staff to switch off lights, IT equipment, etc. One participant considered that having a system "where you can document these, see the impact and be able to communicate it" is very useful (CAT-FG1). Another participant agreed that communicating the efforts with users can help to make them more committed with the performance of the building: "they can see that even a small thing (change), you can have some savings" (CAT-FG1). It was also mentioned that the tool can also be useful to seek cooperation among users, for example, when they have done all that they can within the building controls, but people disagree with their thermal comfort. The ability to communicate with decision-makers as well as staff was emphasised by one participant, who explained that the tools have provided them the required technical support to "talk to the top levels, I mean 'we need investment' as well as to be able to talk with the bottom levels, to tell them 'we are doing all what we can, but the weather is cold or hot'" (CAT-FG1).

#### 4.2.2. Building Communities

Awareness raising and engagement campaigns are run in several ways across the public authorities. For example, primary and secondary schools can have eco-warriors teams or an eco-ambassador as part of the Eco-Schools programme in Leicester (LE1, LE2), "Green Impact" teams comprise environmental champions among staff in different departments of the university (LE3, LE4) and teachers embedding energy awareness in the curriculum of schools in the city of Nuremberg (NUR2).

In Nuremberg, the eco-teacher explained that the school delivers environmental education as part of the curriculum, with the intention that pupils integrate the environmental lessons in their daily lives outside school (NUR2). The interviewee commented that "sometimes there are very aware pupils that tell us [teachers] small things on how we should control energy within the school" (NUR2). In the same interview, the delivery manager (who assisted with the translation) mentioned an example where energy awareness was diffused from schools to homes: "I was told that in one of our schools, some parents complained because the children started switching off lights and other things in their homes."

The ability to communicate with the central energy management team was highly appreciated by some user groups. The eco-teachers created a category especially for the KEiM programme in the EDI-Net online forum (CS9) (Table 4). This category was set up to exchange information about the energy performance and potential energy efficiency interventions in their schools such as insulation and heating in classrooms.

**Table 4.** Case studies related to user engagement.

| Case study | Public Authority | Buildings Involved | Behavioural Interventions |
|---|---|---|---|
| CS9: Eco-teachers in schools | Nuremberg | 12 | • Online forum KEiM category to improve knowledge on energy efficiency |
| CS10: Engaging schools and league table competition | Leicester | 40 | • Staff training<br>• Students engagement<br>• League table competition |

In Leicester, the engagement of EDI-Net in schools was supported with face to face training sessions, regular teacher and business manager meetings, information to schools' governors, regular newsletters via email to school leaders and face to face meetings with premises officers (CS10) (Table 4). Through the tools and training, users have been empowered to understand their consumption. School leaders can review their consumption and compare to other schools. Teachers can easily show the school's energy performance with students, parents and the wider school community. Students use the dashboard to monitor changes that have taken place in the school on a daily basis, and to display the smiley faces on their Eco-boards to raise awareness of energy and water savings. These case studies in Leicester and Nuremberg demonstrate the ability of the tools to increase knowledge and understanding of energy efficiency by different communities inside and outside the school settings.

#### 4.2.3. Energy Competitions

Comparison of energy performance with other buildings can enhance social norms when users perceive competition among buildings and through the interaction of those who do change behaviour. One interviewee found useful the comparison with other schools and buildings in the league tables (LE1). The interviewee thought that comparing with the school "across the park" is useful as the buildings are identical in terms of number of pupils, number of classrooms and the energy usage should be similar. The manager considered that through the comparison, he can understand if his school is conducting appropriate actions. The interviewee believes that showing the comparison of smiley faces to the pupils can help to discuss with them "if they are doing better than us, we can think, well where we are going wrong? ... we should have a smiley face" (LE1). He considered that

competing with schools is useful not only to improve operational practices, but also to engage with the pupils.

Based on the schools' engagement with EDI-Net, Leicester City Council organised a competition over a 3-week period in winter 2018 (CS10) (Table 4). City schools competed to win a banner to be placed on the railings of their school stating they were the 2018 energy saving champions as well as trophies, certificates and other prizes. During the competition, the council's environmental education team was periodically in contact with schools' staff members to update them on their progress and guide them on how to solve consumption problems identified by the dashboard. Staff were eager to learn how energy consumption can be reduced in their schools. This competitive element rather than just monitoring encouraged schools to actively reduce their gas, water and electricity usage.

## 5. Discussion

The EDI-Net systems were welcomed in energy management teams and were found to have a positive impact on energy management operations in several areas. The impact included significant direct financial savings through improved information provision, a contribution to investment decisions and increased engagement from professional stakeholders as well as from building users such as teachers and pupils in schools. The system supports top-down, middle-out and bottom-up energy management processes using metered data as a basis for decision support and communication within and between levels. Each of these modes of operation is discussed below.

### 5.1. Top-Down

The benchmarking tool has been beneficial in supporting energy management investment decisions in the government of Catalonia and the implementation of the Catalan Strategy for Energy Renovation in Buildings. The tool provides an up to date, on demand view into energy management from the perspective of investment and finance. With the support of senior decision-makers, energy saving initiatives are more likely to lead to effective practices by regularly reviewing the ongoing impact of investment in energy management interventions [42]. The EDI-Net system supports a better understanding of the opportunities for behavioural, technical or systemic interventions and more effective communication between the energy management function and senior decision-makers. This ensures that opportunities, uncertainties and constraints are clear to both sides and helps to de-risk the roll-out of proven technologies, reducing organisational and institutional barriers [42,43].

### 5.2. Middle-Out

The EDI-Net service is designed to support central energy managers to monitor energy infrastructure at all scales enabling them to react quickly when issues arise. The system also supports effective communication with stakeholders. Energy professionals typically interact with a variety of stakeholders to manage and optimise energy services across an organisation. They negotiate energy management budgets (upwards) with senior decision-makers in a proactive attempt to upgrade infrastructure. They also work with building users (downwards) and building managers (horizontally) to optimise energy performance locally and minimise waste [13,44].

#### 5.2.1. Technical/Reactive

Operational energy management involves working with buildings and building users to ensure seamless service provision. Opportunities to implement "good housekeeping" must be identified to take full advantage of the "low hanging fruit". Faults that inevitably occur from time to time in building energy systems must be identified and rectified quickly. When overseeing large numbers of buildings, a continuous reactive process of investigation and intervention is required.

The EDI-Net system processes large volumes of complex raw metering data and presents them as simplified summary information highlighting the parts of the data that are most likely to yield actionable information and presenting a standardised, user-friendly interface for navigating the

highlighted dataset. The system is simple to use for both energy professionals and non-expert building users and can be easily shared either by taking screenshots for emailing or embedding in reports or by sharing URLs to the published charts directly in the EDI-Net system.

This mode of operation was clearly observed in several case studies. Most notably in the case studies of the city of Nuremberg related to identifying water wastage. These examples demonstrate very clearly the basic core operational advantage provided by the EDI-Net dashboard. The energy manager needs only to scan quickly over the dashboard in order to be made aware of any issues detected by the automated analysis. In this way, the central energy manager can use the EDI-Net system to participate in a robust feedback loop with the entire building portfolio. Without the EDI-Net system these issues would be detected only after a significant wastage and cost had accrued. Similar benefits of achieving energy and financial savings in public buildings through smart meter monitoring and timely detection of anomalies has been presented by Ferreira and Fleming [45].

### 5.2.2. Strategic/Proactive

In addition to the reactive mode, a more strategic approach is required to maintain the energy infrastructure, implement technological upgrades and to bring about associated efficiency gains in the medium to long term. For example, when new technologies (such as LED lighting or heat pumps) become viable, the energy management function ensures that retrofits are considered as and when appropriate within a specific building or across the portfolio. These may be implemented when existing systems need replacing or are near end of life or when other major work is being undertaken in a building.

A core energy management function involves maintaining a list of preferred technologies and technical specifications and mapping these to an asset register to create a list of potential infrastructural upgrades. Establishing an intervention plan and prioritising investments requires a technical engagement with financial management and senior decision-makers in order to make energy use more salient for senior management and secure funding [44]. It is necessary to estimate the cost of implementation as well as the savings which can be expected. It is also essential to monitor investments to validate the assumptions and improve saving forecasts for future investments.

The EDI-Net service contributes to this function by providing an overview of energy consumption patterns over a longer time scale via the benchmarking tool. Technical interventions are documented and analysed to generate predicted cost/benefit analysis to support the case for investment. Once investments are made, the benchmarking tool can track the impact on usage and validate the return on investment providing decision-makers with increased confidence in the energy management function.

### 5.2.3. Awareness/Engagement

Finally, the central energy management function in large organisations must work with building managers and maintenance personnel working locally to the buildings to ensure building energy systems are operating efficiently and staff are properly trained to optimise their building energy performance. This communication function is extremely challenging when the number of buildings involved is large.

Simple issues such as heating and lighting controls which are set locally can drift from optimal settings and cause significant wastage. Though interventions to fix these issues are effectively zero-cost to implement, they can be difficult to manage at scale since the effort must be distributed across many individuals.

There is evidence that providing the EDI-Net league tables as a public portal into the more detailed energy data is an effective way to engage building managers in monitoring their building performance. This can strengthen the communication between the central energy management function and the distributed energy management capability within an organisation. In the city of Nuremberg, the dashboard and the league tables have been useful tools to increase sensitivity for energy and water

consumption, demonstrated by one school, which had several severe water wastage problems in 2018 which were quickly detected by these tools. The school moved to the top of the schools' league table in 2019. This demonstrates that by using the tools and local knowledge, energy managers can influence building users [44].

*5.3. Bottom-up*

Building users with little or no official responsibility for energy performance of buildings (e.g., 'energy champions') have a critical role in supporting and demanding a robust response to the environmental cost of the energy services they consume. The EDI-Net dashboard provides a feedback system for building users to monitor the impact of any changes they make to their building operation (be they technical or behavioural).

The league table introduces an implicit competitive element which implies that a building community should aspire to perform well relative to its peers in similar buildings. This creates a fertile ground for engagement activities such as competitions with prizes, rewarding individuals or groups towards energy saving attitudes and behavioural change [46]. Each building has a simple public indicator which shows whether the building is performing at, above or below its 'normal' level of usage for the given period and conditions. If there are opportunities to make savings in a building, then it is very easy to climb the list by taking advantage of the opportunities.

The gamification element of the league tables has proven to be highly engaging and a powerful learning tool for schools in Leicester and Nuremberg, which has the potential to bring about a transformative cultural shift towards more deeply engaged building users [47]. For example, engaging with teachers and pupils in schools to support their efforts to mitigate their own environmental impact can have a cumulative effect on a public authority, impacting on policy. Allowing building users to share experience of energy management and collaborate to engage with energy management can ensure interventions achieve maximum impact and are fit for purpose on the ground.

The usefulness and potential of the dashboard for delivering sustainable energy policy has also been realised by the public authorities. For example, the Head of Municipal Energy Management in the city of Nuremberg explained that because several technical energy efficiency measures have already been implemented, engagement with users is becoming more important in the future. This is one of the reasons that the public authority started to show the smiley faces on public screens, and it considers continuing using the dashboard to further develop strategies to engage with the public.

From this perspective, the EDI-Net system goes beyond the traditional energy feedback by enabling engaged stakeholders to evaluate their energy performance and change their energy use behaviour ('practical' feedback), but also to influence policy and decision-making ('policy' feedback) [48].

## 6. Conclusions

The EDI-Net system demonstrates an effective approach for integrating metered utility data into urban management processes via a smart meter monitoring system. When deployed in an organisation, the EDI-Net system automatically processes streams of high-resolution metered utility data from hundreds of buildings. Results are calculated every few hours and the results of analysis are available on demand via a web interface. The system is sensitive to small changes so if a building begins wasting energy, it will be reflected in the interface within hours.

Information is presented in an easy-to-interpret format on a public platform, allowing users to easily absorb and share basic information on related buildings. Expert users can extract further detail from diagnostic visualisations. Strategic managers can view up-to-date financial information, showing the impact of investment in energy management interventions. The system is updated regularly, enabling any engaged stakeholder to participate in a robust feedback loop with the infrastructure at any chosen scale (e.g., a single building, a cluster of buildings or the entire building portfolio).

Central energy managers can monitor the entire building portfolio by regularly checking the system, identifying which (if any) buildings need attention and coordinating a response with relevant

building users. When a problem is detected, experts and engaged building users can begin an investigation to identify and diagnose the problem. Energy managers can implement exploratory actions and see the impact on usage within hours to quickly eliminate possibilities, identify the cause and develop an appropriate response. By providing a common reference point, the system improves communication between central and local experts and between all engaged stakeholders.

With information 'at their fingertips' building users are able to learn how their actions impact on energy performance. It is easy to know when a problem occurs and when the problem has been resolved. Details and can be discussed with experts. Locally, only a few engaged users are needed to feed this information back to the wider community as necessary. Furthermore, the transparent nature of the system and the competitive, gamified element encourages a reduction in the free-rider effect and helps to attribute social value to energy saving behaviours.

The EDI-Net system reduces the transaction costs associated with the identification and diagnosis of the causes of wastage to virtually zero across potentially hundreds of buildings. This has the effect of lowering barriers to action [49], both individually and collectively within communities of building users. Designed with user engagement in mind, the EDI-Net system provides a wide range of users with the opportunity to directly engage with aspects of the infrastructure that were previously inaccessible. Direct access to utility data in formats that are accessible via a simple user interface has been shown to support an improvement in evidence-based decision-making in energy management collectively and at multiple levels. The combination of these improvements across different layers of an organisation has produced a systemic impact that is potentially greater than the sum of the parts.

Evidence of some cultural change (or at least the precursors to a cultural change) was identified through clear pathways for improved access to data, leading to systematic changes in behaviour. For example, operational practice within energy management has changed because a scan of the entire building stock is now possible in a few minutes. This introduces a new challenge since opportunities for swift intervention are presented more often. Therefore, passing this information on to stakeholders local to the building becomes an important strategy for centralised energy management to delegate/devolve the task of investigating the problem and developing a response.

By engaging with local stakeholders with no specific responsibility for energy performance such as teachers and pupils, it is possible to tap into significant enthusiasm for acting locally to support the central energy management function. By giving local stakeholders direct access to the EDI-Net system, these users can conduct their own regular checks on the data as part of local initiatives to keep their own buildings operating efficiently. This aligns well with communities of practice such as Eco-schools in Leicester which reinforce the local interest in data and further distribute the role of energy management across a wider range of stakeholders. There is evidence that the EDI-Net systems have encouraged peer-to-peer learning among building users through friendly competition, creating the conditions for adaptive practices to emerge at the local level.

With more engagement from building users in the energy management process, the centralised energy management function will need to adapt in turn perhaps by refocusing on providing technical support for intervention proposals developed locally. In order to support such a transition, urban analytics tools such as the EDI-Net system need to be ready to adapt to support new and emerging contexts. Integrating systems such as EDI-Net into public authority operations is critical to ensure they reach ambitious national and international greenhouse gas emissions reduction targets. It is key to develop new methodologies and business models, but fundamentally stronger policy frameworks mandating the drive for greater energy efficiency are required.

The evidence presented shows that the introduction of urban analytics systems such as EDI-Net can induce qualitative changes to processes at multiple levels, fostering the vision of smart cities where stakeholders are engaged with the services offered, integrating 'hard infrastructure, social capital including local skills and communities and digital technologies' [50]. The integration of the EDI-Net system points towards a more distributed energy management function. We have observed a shifting of the dynamics to support meso-level communities of practice around clusters of buildings.

The EDI-Net system demonstrates the information hierarchy [51] in action. We see a clear transition from data to information, through knowledge and finally wisdom. The innovative design of the data analytics, intuitive visualisations and simple user interface transform raw data into accessible information that can be readily absorbed by system users [52]. This information informs action that supports stakeholder participation in feedback loops in which the information is embodied as knowledge of how the system reacts to different actions. The social impacts of transparency and gamification serve to magnify this effect and build collective knowledge via peer-to-peer learning which creates the conditions for wisdom to emerge. Wisdom, in this case refers to the collective ability of the heterogeneous community of stakeholders to act in a coherent way to predict and avoid wastage and to invest in appropriate technology. The development of wisdom of this kind is a prerequisite for the kind of transition of the energy infrastructure necessary for a sustainable future. Though information systems such as EDI-Net can be easily deployed, the specific details of how these impacts manifest within the wider dynamics of an organisation are difficult to predict and require further study.

**Author Contributions:** G.S. conceptualization, design and development of the data analytics and user interface, EDI-Net dashboard data curation, formal analysis and visualisation, writing—original draft, review and editing; L.O.-M. conceptualization, methodology, investigation and formal analysis of interviews and focus groups, validation of results with participants, writing—original draft, review and editing. All authors have read and agreed to the published version of the manuscript.

**Funding:** The data referred to in this paper were collected in the project titled "The Energy Data Innovation Network; using smart meter data, campaigns and networking to increase the capacity of public authorities to implement sustainable energy policy" funded by the European Commission's H2020 research and innovation programme under grant agreement number 695916.

**Acknowledgments:** We would like to acknowledge and thank the support of our funder, participants of the interviews and focus groups and to all partners of the EDI-Net project: Stadt Nuremberg, Leicester City Council, Generalitat of Catalonia, CIMNE, Climate Alliance and empirica. We would like to particularly thank to Alexander Nordhaus, Lee Jowett, Juan Antonio Bas Villalobos, Prakash Patel, Michael Richardson, Silvia Mata Rodriguez and Jordi Carbonell for providing the best practice case studies of the project presented in this paper.

**Conflicts of Interest:** The authors declare no conflict of interest. The funders had no role in the design of the study; in the collection, analyses, or interpretation of data; in the writing of the manuscript, or in the decision to publish the results.

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
