# Peer review of "Supporting Decentralised Energy Management through Smart Monitoring Systems in Public Authorities†"

_energies, doi:10.3390/en13205398_

Round 1

Reviewer 1 Report

Please limit self-citations to a maximum of 10% (References). Currently, the share of self-citations by one of the co-authors is almost 30%.

The literature review is insufficient. Please refer to the larger number of scientific publications from recent years, which are thematically related. Please pay particular attention to articles in peer-reviewed scientific journals.

Please pay attention to minor mistakes, for example journals should be capitalized with the capital letter "Energy Policy" (page 20).

Please adapt the manuscript to the requirements of the journal, for example References.

Please consider deleting some of the figures that add little to the manuscript, for example Figure 2, Figure 3 and Figure 4.

Author Response

Thanks for the feedback, we have extended the literature review to include over 50 references, many very recent. We have removed and replaced self-references so that our self-reference rate is below 10%. Figures 2 and 4 were deleted, we kept figures 1 and 3 as we considered the images useful for the explanation provided in the subsequent sections. The revised paper has been proof-read and formatting of references has been checked. 

Reviewer 2 Report

The article has two objectives:

  1. The description of the system
  2. The presentation of the demonstrations of the efficiency of the system

In my opinion the article can be improve if the authors include a third objective and its explanation in the discussion section: Why does the system work so well?

In fact, the authors inform about it. At the end of the introduction they explain that the EDI-Net is in essence an improvement of a collective action (p. 4 and bibliography references 12 and 16 ). The problem is that they do not make use of this fundamental explanation in the section of discussion. They provide a lot of evidences about how the system performs, but they do not explain the fundamental reason of the successful.

I am speaking about the “free-rider” problem in collective action. They must explain why the free-raider problem does not arise when the EDI-Net system is active.

Why the energy professionals and senior decision-makers (perspective middle-out) do not block a determinate improvement? Usually, they have personal interests, but the EDI-Net system has the capacity of acting against their particular interests.

I recommend to the authors a detailed reading of Markus Olson (and his followers) theories. Please, read the principal sources. A good comprehension of the “collective action” notions is essential in this case. A collective action is successful when reduce the transaction costs (transaction costs have three broad categories—information costs, negotiation costs, and monitoring costs). EDI-Net system is a fascinating example of reduction of information and, specially, monitoring costs. When this happens the “free-raider” is obligated to reduce the negotiation costs.       

Author Response

Thanks for the great feedback, we have expanded the literature review with many new references including to the work of Olson and we discuss the free-rider effect and transaction costs. We also added text to the conclusions (section 6, paragraph 5 & 6) where we refer to the free-rider effect and transaction costs. 

We conclude that the success of the system relies on its transparent nature and the competitive, gamified element encourages a reduction in the free-rider effect, helps to attribute social value to energy saving behaviours and that the system reduces transaction costs related to information and monitoring.

This has been a very useful way to frame the impact of monitoring systems for energy management. 

Reviewer 3 Report

The paper is well written and the outputs of the EDI-Net project study are well presented.

One minor suggestion: to reformulate the text in the last paragraph of 5.2.1 at page 15: „The energy manager needs only to scan quickly over the dashboard in order to be made aware of any issues detected by the automated analysis.”

Author Response

Thanks for the feedback. We have revised the paper significantly for readability including your suggested edit.

Reviewer 4 Report

The article Supporting decentralised energy management through smart monitoring systems in public authorities presents the results of original empirical research. The subject of the article is embedded in the sciences of management and quality. The authors raised important and current problems related to the management of urban infrastructure, with particular emphasis on monitoring system for building energy and water usage in large, multi-site, public sector. The subject of the article corresponds to the issues of the journal. The title of the article corresponds to its content. The abstract and keywords correspond to the content of the article. The selection of literature is correct and does not raise any objections. The advantage of the article is the reference to current publications in the field of the studied issue.

The starting point of the reviewed article should be to establish the current state of knowledge in the subject matter under study. There is no reference to the basic canon of science, ie to continuity - research should refer to previous results in order to extend, change or classify existing knowledge. Introduction should provide wider context, substantiation of the need of writing and publishing the paper. The goal of the article in the abstract and in the introduction should be the same.

The description of the research methodology requires more detail. The authors attempted to design a research methodology based on triangulation. The literature on research methodology explains that the essence of triangulation is to take a research question from two or more perspectives in order to obtain convergent results. The authors did not explain the type of triangulation used (e.g. data triagulation, method triangulation, location triangulation, etc.). The authors also did not formulate research questions, which seems necessary in the case of using qualitative methods. At the same time, the use of case studies should start with a research question. Research questions can be descriptive, exploratory or explanatory. Moreover, the research question should be embedded in the current state of knowledge and concern the expected results. The research methodology does not describe the case studies as purposefully selected. The authors should describe the selection of cases in more detail and indicate research entities.

The general structure of the reviewed article is correct. The order of the individual subsections and the content presented in them are also correct. But in my opinion, the data presentation is not very clear. The presentation of the results of the interviews and case studies is haotic, in some parts incomprehensible (e.g. in subsection 4.2.1). The text of the article has been divided into a very large number of short subsections, which gives the impression that the thoughts of the author / authors "break", there is a quick transition from one thread to the next. In addition, in my opinion, a paragraph should consist of min. 3 sentences) - a paragraph covering only one sentence is not correct (e.g. on pages: 11, 12 and 13).

Conclusion should contain evaluation and exact description of achieved results provide a confrontation of the achieved results with previously published papers, author’s opinion of established differences, his/her attitude to the results. Also indicate advantages, limitations, and possible applications.

The language of the article corresponds to the correctness criteria used in scientific statements. The language is clear. But I do not feel an expert in assessing the language quality of the article. I suggest having the manuscript proof read and edited before submitting. If English is a second language for the author, please consider having the manuscript proof read and edited before. The required by the organizers format of the article and spelling and punctuation should be thoroughly checked ubmitting.

The article meets the requirements of the scientific text. The advantage of the article is the international context of the research. The authors' use of empirical data from many regions of the European Union means that the subject may be interesting for readers from various countries. The conclusions from the research may be the beginning of research on a larger scale.

Author Response

Thanks for your detailed and useful feedback we have made significant changes to the paper as a result.   

Point 1. Establish the current state of knowledge in the subject matter under study, references to the basic canon of science, i.e. to continuity - research should refer to previous results in order to extend, change or classify existing knowledge 

Response: We extended the literature review to include over 50 references linking to a wide range of literature related to data visualisation, energy feedback, game theory, economics (free-rider effect and transaction costs), complex systems and information theory.  

Point 2. Introduction should [include] substantiation of the need of writing and publishing the paper. The goal of the article in the abstract and in the introduction should be the same.  

Response: We believe that the substantiation of the need of writing the paper is included in its aim, which was modified to include clear exploratory research questions (instead of adding them in the methodology as requested in the next comment). We also revised that the purpose of the paper in the abstract and in the introduction are aligned. 

Point 3. The description of the research methodology requires more detail […]. 

  • Explain the type of triangulation used (e.g. data triagulation, method triangulation, location triangulation, etc.).  
  • Formulate research questions, which seems necessary in the case of using qualitative methods. 
  • Case studies should start with a research question. Research questions can be descriptive, exploratory or explanatory. Moreover, the research question should be embedded in the current state of knowledge and concern the expected results.  
  • The authors should describe the selection of cases in more detail and indicate research entities. 

Response: 

  • We have clarified that the triangulation approach used in the research relates to method triangulation (interviews/focus groups and case studies) as well as data triangulation (data from different stakeholders at different social and geographical context). We added this information briefly in the introductory paragraph of section 3.  
  • We added phrases in the explanation of the interviews/focus groups (section 3.1) and case studies (section 3.2) that links the purpose of using these data collection methods with the aim (and research questions) of the paper. 
  • Case studies were provided by the delivery managers (consortium partners) in each public authority. These were purposefully selected by the delivery managers based on their own perspective of best practice cases. We hope that the explanation added in section 3.2 clarifies the situation. 

Point 4: The presentation of the results of the interviews and case studies is chaotic, in some parts incomprehensible (e.g. in subsection 4.2.1) […]. In addition, in my opinion, a paragraph should consist of min. 3 sentences) - a paragraph covering only one sentence is not correct. 

Response: We appreciate the comment of the reviewer regarding the presentation of results in section 4 (particularly section 4.2.1). The entire section was revised with an attempt of presenting the results in a logical and coherent manner. We ensured that all paragraphs contain at least three sentences. We also proof-read the entire document to ensure readibility.  

Point 5: Conclusion should contain evaluation and exact description of achieved results provide a confrontation of the achieved results with previously published papers, author’s opinion of established differences, his/her attitude to the results. Also indicate advantages, limitations, and possible applications.  

Response: The conclusions have been reworked to present the authors answers to the research questions with a coherent framing of the wider context including reference to literature and pointing to future work.